# Gene Expression Signature Predictive of Neuroendocrine Transformation in Prostate Adenocarcinoma

**DOI:** 10.3390/ijms21031078

**Published:** 2020-02-06

**Authors:** Paola Ostano, Maurizia Mello-Grand, Debora Sesia, Ilaria Gregnanin, Caterina Peraldo-Neia, Francesca Guana, Elena Jachetti, Antonella Farsetti, Giovanna Chiorino

**Affiliations:** 1Laboratory of Cancer Genomics, Fondazione Edo ed Elvo Tempia, via Malta 3, 13900 Biella, Italy; paola.ostano@fondazionetempia.org (P.O.); maurizia.mellogrand@fondazionetempia.org (M.M.-G.); debora.sesia@gmail.com (D.S.); ilaria.gregnanin@fondazionetempia.org (I.G.); caterina.peraldoneia@gmail.com (C.P.-N.); francesca.guana@fondazionetempia.org (F.G.); 2Department of Research, Molecular Immunology Unit, Fondazione IRCCS Istituto Nazionale dei Tumori, 20133 Milan, Italy; Elena.Jachetti@istitutotumori.mi.it; 3National Research Council - Institute of Analysis, Systems and Computer Science –CNR-IASI, 00185 Rome, Italy; antonella.farsetti@cnr.it

**Keywords:** neuroendocrine prostate cancer, neuroendocrine transdifferentiation, prostate cancer, long non coding RNAs, estrogen signaling pathway, predictive gene signature

## Abstract

Neuroendocrine prostate cancer (NEPC) can arise de novo, but much more commonly occurs as a consequence of a selective pressure from androgen deprivation therapy or androgen receptor antagonists used for prostate cancer (PCa) treatment. The process is known as neuroendocrine transdifferentiation. There is little molecular characterization of NEPCs and consequently there is no standard treatment for this kind of tumors, characterized by highly metastases rates and poor survival. For this purpose, we profiled 54 PCa samples with more than 10-years follow-up for gene and miRNA expression. We divided samples into two groups (NE-like vs. AdenoPCa), according to their clinical and molecular features. NE-like tumors were characterized by a neuroendocrine fingerprint made of known neuroendocrine markers and novel molecules, including long non-coding RNAs and components of the estrogen receptor signaling. A gene expression signature able to predict NEPC was built and tested on independently published datasets. This study identified molecular features (protein-coding, long non-coding, and microRNAs), at the time of surgery, that may anticipate the NE transformation process of prostate adenocarcinoma. Our results may contribute to improving the diagnosis and treatment of this subgroup of tumors for which traditional therapy regimens do not show beneficial effects.

## 1. Introduction

Prostate cancer (PCa) is the most frequently diagnosed cancer in men and the second leading cause of cancer death worldwide [1]. The majority of prostate tumors are adenocarcinomas, which derive from the transformation of glandular prostatic epithelial cells and rely on the influence of the androgens-androgen receptor (AR) axis for development and progression [2]. Hormone-sensitive tumors can then progress into castration-resistant prostate cancer (CRPC), a condition that is resistant to conventional anti-androgen therapies. They must be treated with more powerful hormonal drugs (e.g., enzalutamide or abiraterone) [3]. It has been hypothesized that neuroendocrine prostate cancer (NEPC) arises from one or more CRPC-adeno cells, with a divergent clonal evolution mechanism [4], as a result of selective pressure from androgen deprivation therapy or antiandrogens. NEPCs display both neuronal and endocrine features and are highly aggressive; they transiently respond to chemotherapy and have a poor prognosis [5]. The incidence of NEPC has been rapidly increasing due to a more widespread use of potent androgen deprivation therapies, a piece of evidence that is corroborated by studies that revealed that at least 25% of PCa autopsies show signs of NEPC [6]. NEPC is under-recognized but suspected in patients with progressive disease, despite a normal or modestly elevated PSA, and elevated serum markers of NE differentiation [7]. Since the molecular characterization of NEPC is poor, no standard treatment is available [8], leading to the urgent need for robust criteria for NEPC diagnosis and treatment.

One underestimated aspect of prostate cancer progression regards the cross-talk between androgen and estrogen signaling pathways. It has been demonstrated that ESR2 is down-regulated in epithelial cells during development of prostate cancer [9]. ESR1 is up-regulated in tumor cells as well in the tumor microenvironment. Its expression increases during prostate cancer progression [9]. Moreover, there is evidence that ESR1 antagonists can repress prostate cancer tumorigenicity [10,11] and that abiraterone can activate estrogen receptor [12]. Data were also produced demonstrating that prostate cancer cells can use alternative nuclear receptors signaling pathway, such as ESR1 instead of AR signaling, to propagate [13].

This study aimed to identify molecular features, at the time of prostate cancer surgery, able to anticipate the NE transformation process of prostate adenocarcinoma, to improve the diagnosis and treatment of this subgroup of tumors for which traditional therapy regimens do not show beneficial results. We exploited a cohort of 54 prostate cancer samples for which multi-omics analysis was performed on protein-coding, long non-coding and micro RNAs. We deeply investigated the molecular aspects differentiating tumors with the highest levels of expression of Chromogranin A (CHGA), one of the most valuable markers of NEPC [14], together with the worst clinical outcome of the corresponding patients (classified as NE-like tumors), from the others (classified as AdenoPCa tumors). We found that NE-like tumors were characterized by a neuroendocrine fingerprint, composed by known neuroendocrine markers and by novel molecules. In particular, we identified two long non-coding RNAs (HOTAIR and MALAT1) whose expression increased in NE-like when compared to AdenoPCa tumors, and that was concomitantly positively correlated with CHGA.

To our knowledge, the role of estrogen signaling in NEPC has not yet been investigated. Here we showed that the estrogen signaling, and in particular ESR1, increased in NE-like samples from our cohort and in a model of prostate cancer organotypic slice culture (OSC). Data were validated on other publicly available datasets, providing an opportunity for further studies and for clinical intervention.

Finally, we identified a molecular signature able to predict the neuroendocrine transformation of prostate adenocarcinomas in two independent datasets, and that can be exploited to improve the diagnosis and the therapeutic approach for NEPCs.

## 2. Results

### 2.1. Sample Selection

Fifty-four prostate tumor samples were divided into two groups according to their clinical and molecular features. More in detail, tumors from patients who satisfied at least one of the following clinical criteria (i.e., having undergone clinical progression; died for PCa; having received hormonotherapy; tumor positivity for perineural invasion) together with high levels of Chromogranin A (signal intensity higher than the median value across samples, as assessed by gene expression profiling) were classified as neuroendocrine-like tumors (NE-like, *n* = 22). The remaining samples were classified as classical prostate adenocarcinomas (AdenoPCa, *n* = 32) (Figure 1).

The long-lasting follow-up of our cohort (more than 10 years) was exploited to investigate biochemical recurrence and clinical features of NE-like and AdenoPCa samples. The Gleason score was statistically different between the two groups, with higher scores in NE-like tumors, while biochemical relapse (BCR), grade, age at surgery and initial PSA did not show any statistically significant difference (Table 1).

### 2.2. Identification of Differentially Expressed mRNAs, lncRNAs and miRs

Gene and microRNA (miRNA) expression profiling with Agilent 8x60k arrays was performed to identify protein-coding RNAs (pcRNA), long non-coding RNAs (lncRNA) and miRNA differentially expressed in NE-like versus AdenoPCa samples. Data reduction analysis using either whole mRNA or miRNA profiles was not able to divide NE-like from AdenoPCa tumors (Appendix A), pointing to the evidence that the two classes of samples had very similar molecular patterns.

LIMMA package was then used to perform a two-class comparison between NE-like and AdenoPCa expression profiles on pcRNAs, ncRNAs and miRNAs. We decided to initially use weak cut-offs (absolute logFC > 0.378 and *p*-value < 0.05) for the selection of differentially expressed genes, since we were looking even for small differences at the molecular level in two groups of samples that were undistinguishable from a histopathological point of view. Globally, 295 pcRNAs, 24 lncRNAs, and 6 miRNAs were up-regulated, while 206 pcRNAs, 15 lncRNAs, and 1 miRNA were down-regulated in NE-like vs. AdenoPCa samples (Table 2 and Figure 2A–C). No chromosomal enrichment was found for any list of differentially expressed transcripts, as assessed by the Enrichr tool.

Class comparison between NE-like and AdenoPCa tumors revealed that among the most up-regulated genes, there were many neuroendocrine markers, neuropeptides, genes expressed in thyroid and genes related to castration-resistance, as well as the estrogen receptor together with other estrogen-linked genes (Figure 2D). Among long non-coding RNAs, HOTAIR and MALAT1 appeared more expressed in NE-like tumors in a statistically significant manner (Appendix A). 

On the contrary, genes whose loss/down-regulation was implicated in prostate cancer cell invasion, metastasis formation and worse prognosis (AR, IGF1, IGF2, TGFB3, SEMA3E) were down-regulated in NE-like versus AdenoPCa tumors (Figure 2D).

### 2.3. Functional Enrichment Analysis

David and Metacore tools were used to perform functional classification on the lists of differentially expressed genes in NE-like vs. AdenoPCa samples in terms of gene ontology, pathway maps, molecular networks, diseases, and drug targets. Interestingly, Metacore analysis revealed that many elements related to the estrogen pathway emerged from the study of up-regulated genes (Figure 3A).

ESR1 was more expressed in NE-like tumors when compared to AdenoPCa samples (logFC = 0.6, pv = 0.03), together with another estrogen receptor gene, GPER1 (logFC = 0.39, pv = 0.047), while ESR2 did not show changes in expression (Appendix A). Although the increase of gene transcription of ESR1 was rather small, the average absolute ESR1 expression in the samples from our cohort was 9.6 (expressed as log2 intensity values). If we consider that the range of expression for all the probes present on the arrays was the following: I quartile = 6.6, median = 7.6, III quartile = 9.96, we can affirm that ESR1 was expressed at high levels (near to the 75° percentile). 

Also diseases such as neuroectodermal tumors and prostatic neoplasms showed statistically significant enrichment for up-regulated genes. On the other hand, the androgen receptor signaling cross-talk network was enriched among down-regulated genes (Figure 3A).

Functional analysis of up-regulated genes in NE-like vs. AdenoPCa samples using David showed enrichment in biological processes typically controlled by neuropeptides, such as modulation of ion currents, synaptic vesicles and catecholamines secretion. Endocrine features were also present (thyroid gland development, regulation of hormone levels), together with positive control of ERBB/EGFR pathway (Figure 3A). Concerning biological processes over-represented among down-regulated genes, there were DNA replication, growth (NE prostate cells have a lower rate of proliferation), fatty acid/lipid oxidation, xenobiotic metabolic process, IGF-1 pathway, response to hypoxia, neuron death (Figure 3A).

Network analysis based on manually curated information from the literature and with differentially expressed pcRNAs, ncRNAs and miRNAs, performed by using Metacore, identified direct interactions between different molecular features used as input. Twenty-four hubs with 5 or more connections were identified. ESR1 and AR were the most connected proteins in the network, with 88 and 82 edges, respectively (Table 3 and Figure 3B), and with an overlapping subset of genes regulated by both receptors, together with specific ones (Appendix A).

Transcription factor interactome analysis pointed out several enriched transcription factors that may putatively control a significant number of up- or down-regulated genes in the NE-like vs. AdenoPCa comparison. Again, ESR1 and AR were among the most statistically significant transcription factors for up-regulated genes, while ESR2 and AR for down-regulated genes (Figure 3C). This results reflected the ability of AR to act both as a transcriptional activator or repressor, depending on specific DNA binding elements or specific coactivator and corepressor protein complexes.

Finally, the drug target identification function, available within Metacore, was able to identify 14 known up-regulated and 8 known down-regulated drug target genes among the list of DEGs (Table 4).

Pre-ranked Gene set enrichment analysis (GSEA) unraveled a statistically significant association between up-regulated genes in NE-like tumors and up-regulated genes of the late “NE tumor” gene signature by Mauri et al. [15] and the CRPC-NE vs. CRPC-Adeno tumors comparison by Beltran et al. [4] (Appendix A).

### 2.4. Estrogen Pathway in Neuroendocrine Transdifferentiation of Prostate Cancer

Estrogen signaling emerged as over-represented in our cohort of NE-like tumors compared to AdenoPCas, suggesting a possible role in the neuroendocrine transdifferantiation process of prostate adenocarcinomas. We decided to investigate this possibility more in detail by exploiting publicly available gene expression data. The GEO series GSE59984 [16] consisted of a dataset of a patient-derived xenograft (PDX) model of NEPC developed from a hormone-naive prostate adenocarcinoma through castration. In the last two samples of this transdifferentaition process (LTL331_Cx_relapsed and LTL331R), classified as neuroendocrine by the authors, ESR1 expression levels increased if compared to the remaining 12 adenocarcinomas (logFC = 1.14 and Adj.P.Val = 1.14 × 10^−6^), with a positive correlation with CHGA and a negative correlation with AR (Figure 4A–D).

To investigate whether prostate cancer displayed NE features when the activation of the estrogen pathway was present, we performed gene expression profiling on a model of prostate cancer organotypic slice culture (OSC) treated with estradiol (E2), which showed an up-regulation of elements related to neuronal and endocrine functions (Appendix A). 

We then exploited VCaP cells from the GSE43988 GEO series [13], where ESR1 expression was induced. CHGA was up-regulated in VCaP ERa vs. control cells, together with genes involved in neural and hormonal processes (Appendix A). 

Similar results were obtained by analyzing prostasphere-forming cells of human LNCaP cells treated with estradiol (GSE37531): also in this case, we noticed an overrepresentation of processes and pathways related to neuroendocrine functions (Appendix A). 

Interestingly, for all the three independent datasets analyzed, an enrichment of up- and down-regulated transcriptional targets for both AR and ESR1 emerged, suggesting a possible interplay between these two proteins in controlling the neuroendocrine transdifferentiation process of prostate cancer (Appendix A).

To test the hypothesis that estrogen signaling somehow participates in neuroendocrine prostate tumor transdifferentiation, while androgen signaling is suppressed, we analyzed the overlap between i) up-regulated genes in NE-like vs. AdenoPCa, ii) up-regulated genes in PCa OSC treated with estradiol (E2) and iii) down-regulated genes in PCa OSC treated with dihydrotestosterone (DHT) (Figure 4E). Chromogranin A was concomitantly overexpressed in NE-like tumors and in E2-treated OSC and down-regulated in DHT-treated OSC, together with LAMP3 and SERPINB11, two genes overexpressed in PCa and involved in cellular transformation and invasiveness. Genes involved in prostate cancer invasiveness, advanced stages, and progression to metastatic disease such as PSCA, PTK6, S100B, TMPRSS4, were found up-regulated in NE-like tumors and in E2-treated OSC, together with TFF1, one of the major estrogen-regulated proteins and an indicator of estrogen receptor functionality. On the contrary, CDK19 and NOS2, overexpressed during prostate cancer progression, were induced in NE-like tumors and down-regulated in DHT-treated OSC (Figure 4E).

### 2.5. LncRNA and miRNA Expression Profiling

Among the list of differentially expressed lncRNAs, HOTAIR and MALAT1 emerged as up-regulated in NE-like vs. AdenoPCa tumors (logFC 0.63 and 0.53, *p*-value = 0.0008 and 0.02, respectively) (Appendix A). HOTAIR was also up-regulated in NEPC vs. PCA (adjusted *p*-value = 0.00057) in the study from Beltran and colleagues [8], while MALAT1 was not present in the list of differentially expressed genes. To give strength to our finding, we exploited the PDX model of NEPC from the GEO series GSE59984. The experiments were performed on Agilent SurePrint G3 Human GE 8x60K, a platform array comprehensive of a high number of long non coding RNAs. Here, both HOTAIR and MALAT1 were up-regulated (logFC = 0.64 and logFC = 0.75, respectively). In particular, HOTAIR was differentially expressed in a statistically significant manner (adjusted *p*-value = 4.37 × 10^−5^).

Interestingly, HOTAIR expression was positively correlated with CHGA and other neuroendocrine and CRPC markers and negatively correlated with AR and other genes involved in prostate carcinogenesis (Figure 5A,B). Globally, genes positively correlated with HOTAIR (Pearson score > 0.3 and *p*-value < 0.05) were involved in processes such as ion transport, regulation of hormone levels, vesicle organization, secretion, hormone metabolic process and cell death (enrichment *p*-value < 0.05, data not shown). On the other hand, genes negatively correlated with HOTAIR (Pearson score < −0.3 and *p*-value < 0.05) were enriched in terms of metabolic/catabolic processes, mRNA processing, cell cycle, response to stress, autophagy, IGF-1 Signaling Pathway, Prostate cancer, HIF-1 signaling pathway, Estrogen signaling pathway (enrichment *p*-value < 0.05, data not shown). We exploited the external dataset GSE61270 [17], in which lentivirus-infected control or HOTAIR stable LNCaP Cells were hormone-deprived for 3 days, and then treated with ethanol or synthetic androgen (1 nM R1881 for 24 h). Also in this case, an overlap was found between genes induced by HOTAIR and genes overexpressed in NE-like tumors, such as prostate prognostic markers (PSCA, FOLH1, TRIB1, TACSTD2), neuronal and hormone-related genes (DDC, ESR1, SCG2) and androgen-responsive genes (AGR2, GPC3) (Figure 5C). 

MiRNA analysis led to a few deregulated miRNAs: 6 up-regulated (miR-31-5p, miR-200a-3p, miR-200b-3p and miR-429, miR-21-3p, miR-1268a) and 1 down-regulated (miR-99a-5p) in NE-like vs. AdenoPCa comparison. Interestingly, miR-31-5p was the miRNA showing the most negative correlation with AR (Pearson score −0.43 and *p*-value = 0.0026) and concomitant positive correlation with ESR1 (Pearson score 0.49 and *p*-value = 0.0004) in our cohort of prostate cancer tissues. Since miRNAs and lncRNAs mainly orchestrate precise gene regulatory networks, we decided to investigate more in detail the connections existing between AR, ESR1, HOTAIR and miR-31-5p. A direct interaction network, based on scientific evidence, emerged between the four molecules, with a mutual regulatory model between miR-31, which inhibits the expression of AR [18], and AR itself which in turn can repress microRNA 31 transcription [18]. HOTAIR can enhance both AR and ESR1 by an activating binding mechanism [17,19], while AR transcriptionally inhibits HOTAIR [17]. Discordant effects of ESR1 on HOTAIR and AR have been reported in the literature [19,20,21,22], while AR is known to inhibit ESR1 in tissues and cell lines of different origin [23,24] (Figure 5D).

### 2.6. Gene Signature of Neuroendocrine Prostate Cancer

The list of differentially expressed pcRNAs and ncRNAs (both up- and down-regulated) in NE-like vs. AdenoPCa samples of our cohort was firstly used for unsupervised hierarchical clustering on RNA-Seq data from Beltran [4] consisting of 15 CRPC-NE and 34 CRPC-Adeno samples. Only 2 NE samples were misclassified according to clustering analysis (Figure 6A). 

To identify a gene signature able to predict the neuroendocrine transformation of prostate adenocarcinomas, penalized logistic regression was applied to our dataset, using Ridge penalty and starting from 8 transcripts overexpressed in NE-like vs. AdenoPCa from our cohort, in Mauri’s Late NEPC [15] and in GSE59984 PDX models of NEPC [16]. These genes were SLC35D3, CRABP1, CHGA, SCG2, NEFM, SYT4, NOL4, SMPD3. We built a score by summing the Ridge coefficient estimates of the variables multiplied by their values and tested the score both on Beltran dataset [4] (AUC = 0.871) and on GSE66187 dataset, containing four histologically confirmed NEPC and 20 AdenoPCa [25] (AUC = 1) (Figure 6B,C, respectively).

## 3. Discussion

Neuroendocrine transdifferentiation of prostate cancer is still poorly understood and is thought to affect approximately 25% of the tumors of men who die of prostate cancer [26]. This transformation leads to a neuroendocrine phenotype, similar to de novo neuroendocrine prostate cancer, which is associated with low AR signaling and poor prognosis. With the recent introduction of new generation potent AR pathway inhibitors, such as abiraterone and enzalutamide, an increasing number of prostate tumors acquire NEPC features and transdifferentiates in the so-called “therapy emergent NEPC”. Our knowledge of NEPC biology is still incomplete and there is an urgent need to develop effective treatments for this particularly aggressive tumor that is resistant to conventional therapies. Furthermore, since not all tumors are prone to develop such a transformation, it would be essential to know whether patients destined to develop NEPC have molecular features at baseline that predispose them to therapy emergent NEPC after androgen deprivation therapy [27]. This knowledge will allow us to adopt in advance alternative strategies for treatment in the attempt of prolonging progression-free survival. In fact, as Wang and collaborators highlighted in a pooled analysis of published data [28], the onset of NEPC is associated with a shorter time to progression under ADT.

We analyzed the expression profiles of coding, long non-coding and microRNAs in a cohort of freshly frozen localized prostate tumors from patients who underwent radical prostatectomy between 2003 and 2005 and for whom we have a follow-up of more than ten years. By coupling Chromogranin A mRNA expression with the response to hormonal therapy, survival, and perineural invasion information, samples were divided into NE-like and AdenoPCa. Multi-omics comparison of the two groups revealed that NE-like tumors were characterized by the overexpression of a neuroendocrine fingerprint, composed by known neuroendocrine, neuronal and CRPC markers, and novel molecules, including long non-coding RNAs such as MALAT1, as well as the estrogen receptor signaling. We found an up-regulation of HOTAIR in NE-like samples, an evidence recently corroborated by two independent studies [29,30]. Positive regulation of the ERBB/EGFR pathway was also highlighted, and this finding is supported by the fact that this pathway comes into play in the sustainment of androgen-independent prostate cancers [31]. Gene expression profile studies confirmed our results on independent NEPC samples from patients or animal models, and on organotypic slice cultures from fresh human tumors or cell lines that were perturbed with specific treatments such as estradiol, DHT or HOTAIR overexpression. Among the most significantly enriched pathways within transcripts down-regulated in NE-like tumors, we found the androgen receptor signaling cross-talk. Kelly and Balk [32] postulated that PCa cells with reduced AR might have survival advantages once their growth becomes AR-independent after AR potent inhibitor (ARPI) treatment. In NE-like vs. AdenoPCa tumors, we also saw up-regulation of some AR targets, among which HOTAIR and ESR1 that are usually inhibited by AR. Furthermore, transcription factor enrichment analysis revealed that ESR1 and AR are among the most statistically significant transcription factors controlling up-regulated genes, suggesting the hypothesis of alternative receptor signaling in NEPC. In fact, genes regulated by both receptors partially overlapped. AR and ESR1 can both act as transcriptional activators or repressors, depending on specific DNA binding elements or specific coactivator and corepressor protein complexes. Common targets down-regulated in NE-like prostate tumors included LDHA, already described in prostate cancer cells and activated by AR [33]. No studies in PCa were available on the interaction between LDHA and ESR1. Interestingly, among the common up-regulated targets, UGT2B15 and HOTAIR were described to be inhibited by AR in prostate tumors [17,34] and activated by ESR1 [20,35]. AR specific targets were mainly involved in sulfur compound biosynthetic process, response to lipid, response to steroid hormone, regulation of ion homeostasis, signal transduction, epithelial cell development. The majority of them has already been described in PCa and most of the down-regulated ones in NE-like tumors were indeed activated by AR. ESR1 specific targets were mainly involved in T-helper cell differentiation, regulation of hydrolase activity, response to hormone, negative regulation of cell proliferation, regulation of neuron death, cell differentiation. Among ESR1 specific targets that were up-regulated in NE-like tumors, only TFF1 has already been studied in PCa, where it was activated by the estrogen receptor [36].

The function of ESR1 in NEPC has not yet been investigated, while its role on prostate tissue is ascertained. For this reason, there is an increasing interest in the use of SERMs (selective estrogen receptor modulators) as a potential singular or adjunctive agents for the treatment of PCa [37], with many ongoing clinical trials. Raloxifene, a mixed estrogen agonist/antagonist, has demonstrated some favorable results, particularly in androgen-independent CRPC [38,39,40]. On the contrary, Fulvestrant, a pure estrogen receptor antagonist, did not show any effect on median time to progression, or median overall survival [41]. Despite some encouraging results, SERMs are not currently used in PCa treatment. Concerning ESR1 protein abundance, there is a lack of NEPC samples in public databases. In one study, SILAC-based quantitative proteomic profiling was performed on prostate cancer progression to androgen independence. ESR1 protein was not found overexpressed but induced candidates had central nodes in the ESR1 signaling cascade [42]. In another study, ESR1 was among candidates for drug repurposing against the “BM2” metastatic PCa subtype in which, for a fraction of tumors, the expression of neuroendocrine markers such as CHGA by mass spectrometry was found [43]. Our results on a putative role of ESR1 signaling pathway in the neuroendocrine transdifferentiation process of prostate cancer may provide an opportunity for further studies and for clinical intervention, with androgen deprivation therapy coupled with estrogen blockage on the basis of individual molecular features, in a personalized medicine point of view.

Finally, to identify an NEPC expression signature able to predict NE transdifferentiation in localized prostate cancer at the time of diagnosis, we exploited expression profiles of several independent cohorts of human tumors or mouse xenografts and trained a classifier on our dataset by fitting a logistic regression model. The predictive score was then tested both on Beltran dataset and on patient-derived xenograft tumors, where the development of NEPC from conventional prostatic adenocarcinoma was correctly predicted. Our classifier of course included CHGA, an unquestioned NEPC marker. It also contained SMPD3 that plays a role in the metastatic bone formation and has been found deregulated in bone-forming osteoblasts upon treatment with osteolytic prostate cancer cell-conditioned medium [44]. SCG2 encodes for secretogranin, among the NE markers identified in prostate small cell carcinomas by a molecular characterization study that highlighted a diverse repertoire of genes reflecting characteristics of their NE cell of origin [45]. CRABP1 has a pro-tumorigenic and a pro-metastatic activity in mesenchymal and neuroendocrine tumors [46]. SLC35D3, which showed a dramatic up-regulation also in NEPC PDXs from the independent dataset GSE59984 (data not shown), is specifically expressed in the substantia nigra, striatum, and olfactory bulb in mouse brain and it functions as an ER chaperone for dopamine receptor D1 trafficking to the plasma membrane [47]. Finally, it has been demonstrated that NEFM was diffusely expressed in prostate cancer cells undergoing neuronal trans-differentiation [48] and NOL4 was described as a biomarker for aggressiveness prostate cancer [49]. In our study, we derived a NEPC signature starting from primary prostate adenocarcinomas, as assessed by an experienced pathologist. On the other hand, studies from other groups used metastases from neuroendocrine transdifferentiated prostate cancer, treatment-emergent small cell neuroendocrine prostate cancer [4,50], or pure NE tumors [8]. In the study by Beltran and colleague, which represents the existing dataset with the largest number of NE transdifferentiated prostate samples, a molecular classifier composed by 70 genes was developed by exploiting expression data of genes prioritized by genomic, transcriptomic or epigenomic status [4]. Concerning the methodologies used to derive gene signatures in bioinformatics studies, Tsai and colleagues started from published datasets including adenocarcinomas with NE differentiation, small cell prostate carcinomas, and large cell neuroendocrine carcinomas, using an outlier-based meta-analysis approach [51]. Aggarwal and colleagues developed a gene expression signature of t-SCNC, with high internal accuracy on leave-pair-out cross-validation that was applied to three external mCRPC data sets and correctly categorized neuroendocrine tumors. No detailed information on the methodology was available [50]. In the study by Cheng et al., clustering and differential expression analysis was performed to derive a NEPCa signature [52]. Alshalalfa and colleagues [53] applied a logistic regression model with ridge penalty, a more appropriate method to build a classifier that was also used in our study. They derived a 212-gene signature to identify treatment-naïve primary prostatic tumors that were molecularly analogous to prostatic small cell neuroendocrine carcinoma. 4 genes were in common between our 8-gene signature and Alshalalfa’s 212-gene signature (i.e. SLC35D3, NEFM, SYT4, and SMPD3). We believe that our modeling approach that starts from primary prostate adenocarcinomas with 10 years of follow-up, combines results derived from other studies to build a robust classifier and then validates the classifier on neuroendocrine transdifferentiated tumors, adds novelty in this field and yields to robust results.

In conclusion, a score that combines the expression of eight transcripts deregulated in several NEPC was derived and could be applied to predict NEPC transdifferentiation in localized tumors, at the time of diagnosis. These results may be complemented with other tools that are in progress to achieve sensitive and earlier detections, such as serum chromogranin A detection, analysis of NEPC-specific circulating tumor cells and of circulating tumor DNA [54,55].

## 4. Materials and Methods

### 4.1. Prostate Samples Recruitment

Prostate cancer tissue samples containing at least 70% neoplastic cells were obtained from 54 patients who had undergone radical prostatectomy at the Hospital of Biella (Italy). All patients gave informed consent and the study was approved by the local Ethical Committee (study ID: SERPROS C.E. 149/11).

### 4.2. Organotypic Slice Culture (OSC) Preparation and Treatment

Organotypic slice cultures (OSCs) were generated as described in Aiello et al. (SREP 2016). Medium was replaced daily and after 3/4 days slices were treated with or without 17β-estradiol (E_2_, 10^−7^M) for 6 hours before harvesting. This study was authorized by the ethical committee of Fondazione Policlinico Gemelli-Università Cattolica of Rome, Italy (Protocol number: 25519/16; ID:1247). All procedures were conducted according to the principles expressed in the Declaration of Helsinki, the institutional regulation and Italian laws and guidelines.

### 4.3. Gene Expression Analysis 

For gene expression profiling of prostate cancer samples, RNA isolated from 54 prostate cancer tissues and from a commercial pool of RNA from organ donor healthy prostates (Becton Dickinson, Franklin Lakes, NJ, USA) was amplified using the Amino Allyl MessageAmp aRNA Kit (Life Technologies, Carlsbad, CA, USA) to obtain amino allyl antisense RNA (aaRNA) following the method developed by Eberwine and coworkers. Briefly: mRNA was reverse transcribed in cDNA single strand; after the second strand synthesis, cDNA was in vitro transcribed in aaRNA including amino allyl modified nucleotides (aaUTP). Both dsDNA and aaRNA underwent a purification step using columns provided with the kit. Labeling was performed using NHS ester Cy3 or Cy5 dyes (Amersham Biosciences, Little Chalfont, UK), able to react with the modified RNA. Labeling efficiency and mRNA quantity were checked using NanoDrop ND-1000 Spectrophotometer (Thermo Scientific, Waltham, MA, USA).

Equal amounts of differentially labeled specimens from sample and reference were put together, fragmented and hybridized to oligonucleotide glass arrays with sequences representing 27,958 Entrez Gene RNAs and 7,419 lincRNAs (Human Gene Expression 8x60K Microarray, Agilent Technologies, Santa Clara, CA, USA). All steps were performed using the In Situ Hybridization kit-plus (Agilent Technologies) and following the 60-mer oligo microarray processing protocol (Agilent Technologies). Then, slides were washed following the Agilent wash procedure. Slides were scanned with the dual-laser Agilent scanner G2505C. For each sample, a dye-swap replicate was performed. Raw data elaboration was carried out with Bioconductor using R statistical language. The LIMMA (LInear Models for Microarray Analysis) package was used to analyze gene expression profiles of prostate cancer tissues. Raw intensity values were background subtracted (method = normexp, offset = 50) and normalized using the loess method, for the within array normalization, and Aquantile, for the between array normalization. Duplicated probes were averaged. Then, Cy3 and Cy5 channels were separated using the separate channel analysis of two-color data, available within the Limma package. 

For gene expression profiling of OSCs, RNA quality and quantity were assessed using Agilent 2100 bioanalyzer (Agilent Technologies) and NanoDrop ND-1000 spectro-photometer (Thermo Fisher Scientific, Waltham, Massachusetts, USA), respectively. Gene expression profiling was carried out using the two-color labeling method by means of Low Input Quick Amp Labeling Kit (Agilent Technologies): labeling, hybridization, slide washing and scanning were performed following the manufacturer’s protocols. Briefly, mRNA from 100 ng of totRNA was amplified, labeled with Cy3 or Cy5 and purified with columns. 300 ng of differentially labeled specimens were hybridized on Agilent Human Gene Expression v3 8x60K microarrays. Then, slides were washed and scanned using the Agilent Scanner version C (G2505C, Agilent Technologies).

miRNA expression profiling was carried out on primary prostate tumors (*n* = 48) using the 1-color labeling method and following the manufacturer’s protocols (Agilent Technologies). Briefly, 100 ng of total RNA was dephosphorylated and denatured; then a ligation and labeling step with Cy3 was performed. Samples were hybridized to oligonucleotide glass arrays (Agilent Human miRNA Microarray 8x15K, V3) with sequences representing probes for 866 human and 89 human viral miRNAs from the Sanger database version 12.0 (Human miRNA Microarray V3, Agilent Technologies). Slides were scanned with the dual-laser Agilent scanner G2505C. Raw intensity values were background subtracted (method = normexp, offset = 20) and normalized using the quantile method using the limma package.

pcRNAs, lncRNAs and miRNAs with log fold change higher than 0.378 or lower than −0.378 and *p*-value < 0.05 were considered to be differentially expressed.

Raw and processed data were deposited on the GEO Omnibus database (GSE60329, GSE142288 and GSE142287). 

### 4.4. Cluster and Principal Component Analysis

MeV version 4.9.0 was used for unsupervised hierarchical clustering performed on the global gene expression profiling from Beltran dataset. Pearson distance as similarity metrics and average linkage as the linkage method were used.

The principal component analysis was performed on the whole gene and miRNA expression data with the *prcomp* function, available within the *stats* package in R.

### 4.5. Functional Analysis

Over-represented biological processes, molecular functions, and cellular components of the Gene Ontology (GO) were investigated with the functional annotation tool available within DAVID 6.8 (https://david.ncifcrf.gov/). 

MetaCore version 19.4 (Clarivate Analytics, Filadelfia, PA, USA) was used for network and pathway analysis for transcription factor interactome analysis and for drug target identification. In particular, a network consisting of direct interactions was built between AR and ESR1 and modulated pcRNAs, lncRNAs, and miRNAs. 

Enrichr was used to test if particular chromosomal locations were enriched in the group of DEGs in NE-like vs. AdenoPCa samples [56]. J-Circos was used to perform circular plot with human chromosomes [57].

### 4.6. Gene Set Enrichment Analysis (GSEA)

Pre-ranked GSEA was used to evaluate significant enrichment in predefined sets of genes. 

### 4.7. External Datasets

The following datasets were downloaded from the Gene Expression Omnibus (GEO) database [58]: GSE5998, GSE4398, GSE6127, GSE5998, GSE3753, GSE6990, and GSE66187. All the datasets except one (GSE43988) were from microarray experiments and were analyzed by the limma package. For GSE43988 on RNA-Seq data, RPKM values produced by the authors were used.

Beltran’s dataset was downloaded from cBioPortal website [59] as Z-score values.

### 4.8. Statistical Analysis

Fisher’s exact test was used to perform the statistical comparison of proportions, using the software language R. The *cor.test* function, available within the R *stats* package, was used to assess linear dependence between variables through Pearson correlation.

### 4.9. NEPC Classifier

To build a classifier able to discriminate AdenoPCa from NEPC in Beltran’s dataset, a penalized estimation was used, starting from 8 transcripts that were differentially expressed in NE-like vs. AdenoPCa from our cohort, in Mauri’s Late NEPC and in GSE59984 PDX models of NEPC. In particular, we fitted a logistic regression model with the Ridge penalty on the 8 selected genes, using the *glmnet* R package. The best tuning parameter was chosen by ten-fold cross-validation. Then, we built a score by summing the Ridge coefficient estimates of the variables multiplied by their values. The area measured the accuracy of the classifier was measured by the area under the ROC curve (AUC). 

## 5. Conclusions

Our data contributed to describing molecular features of localized prostate tumors, at the time of surgery, that may anticipate neuroendocrine transformation of prostate adenocarcinomas. This was achieved by expression profiling analysis of coding, long non-coding and microRNAs in a cohort of freshly frozen localized PCas from patients with over 10 years of follow-up. In particular, the subgroup of PCas with highest levels of Chromogranin A and worst clinical outcome and/or perineural invasion at the time of diagnosis showed overexpression of a neuroendocrine fingerprint, composed by known neuroendocrine, neuronal and CRPC markers, and novel molecules, including long non-coding RNAs such as MALAT1. Interestingly, HOTAIR and ESR1 were overexpressed and AR underexpressed in NE-like samples, evidence that was also validated on independent datasets. Finally, we built a score by combining the expression of eight transcripts derived by our and two independent studies and applied it to predict NEPC transdifferentiation on two other published cohorts. These results provide an opportunity for further research, clinical intervention, and early detection of localized prostate adenocarcinomas that are prone to develop neuroendocrine transdifferentiation.

## Figures and Tables

**Figure 1 ijms-21-01078-f001:**
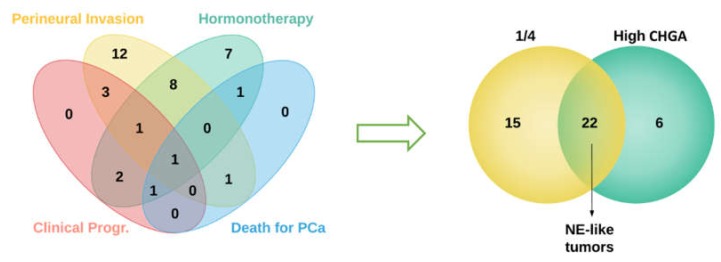
A Venn diagram showing the intersection between different clinical features of patients with prostate adenocarcinomas (left panel). Tumors from patients who satisfied at least one out four clinical criteria (indicated as 1/4 in the right panel) were intersected with tumors with high levels of Chromogranin A expression (right panel).

**Figure 2 ijms-21-01078-f002:**
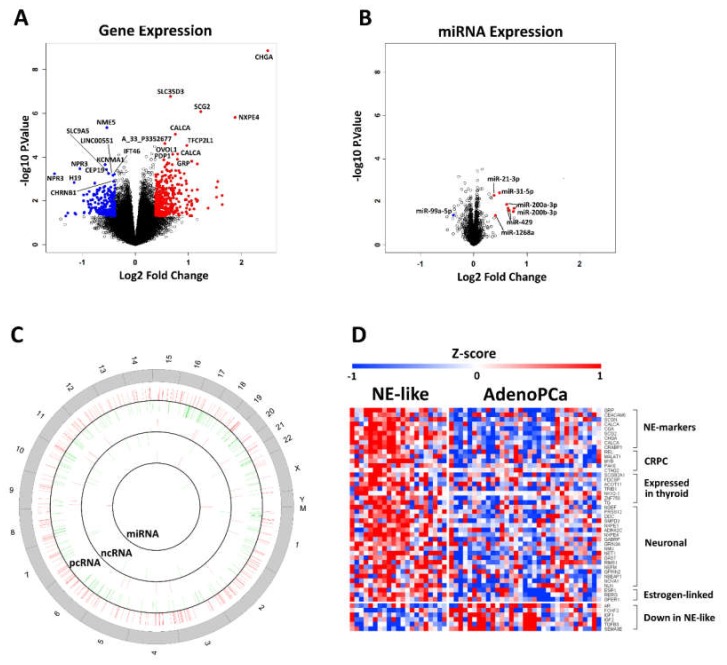
Volcano plots visualizing differential expressed genes (**A**) and miRs (**B**) obtained by class comparison analysis. Blue and red dots represent down- or up-regulated features, respectively. Symbols for the top 10 up- and down-regulated genes and miRNAs are displayed. (**C**) A circular plot of the human genome divided into three concentric circles, each representing differential expression at different molecular levels (pcRNA, ncRNA, miRNA). Red and green bars represent the position of each up- or down-regulated molecular feature along chromosomes, respectively. (**D**) Expression profile of the genes involved in NE prostate cancer induction and maintenance and concomitantly up-regulated in NE-like vs. AdenoPCa samples (upper part of the heatmap) and of the genes whose loss is implicated in prostate cancer invasion, metastasis formation and worse prognosis and concomitantly down-regulated in NE-like vs. AdenoPCa samples (lower part of the heatmap). Z-score values are reported (logIntensities were median-centered and divided by standard deviation).

**Figure 3 ijms-21-01078-f003:**
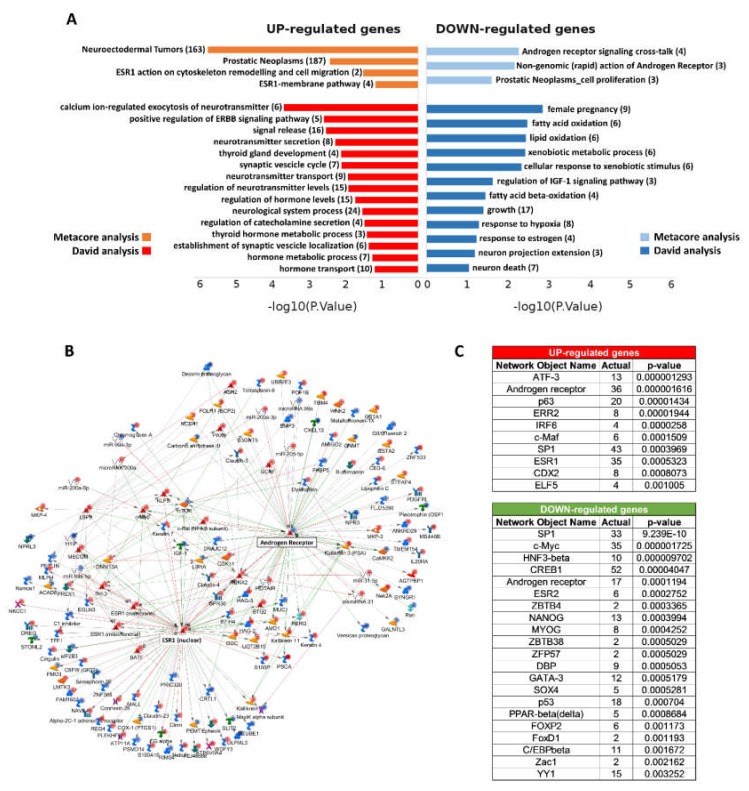
(**A**) A selection of pathways derived from functional enrichment analysis of DEGs using Metacore (orange and light-blue bars) and David analyses (red and blue bars). The negative logarithm of the *p*-value (base 10) is reported on the X-axis. FDR *p*-values range from 3.31 × 10^−5^ to 0.9. The number of genes involved in each process is indicated in brackets. The complete list of pathways and processes is available in Appendix A. (**B**) The network of direct interactions starting from or pointing to AR and ESR1 proteins. More in detail, 70 connections started from AR, and 12 pointed to AR, while 75 links started from ESR1, and 26 pointed to ESR1. The objects that are over- or underexpressed in the data are marked with red/blue circles, respectively. Different edge colors were used to represent the activation (green), inhibition (red), and unspecified (grey) effects. (**C**) Transcription factor interactome analysis pointed out several enriched transcription factors that may putatively control a significant number of up- or down-regulated genes in the NE-live vs. AdenoPCa comparison.

**Figure 4 ijms-21-01078-f004:**
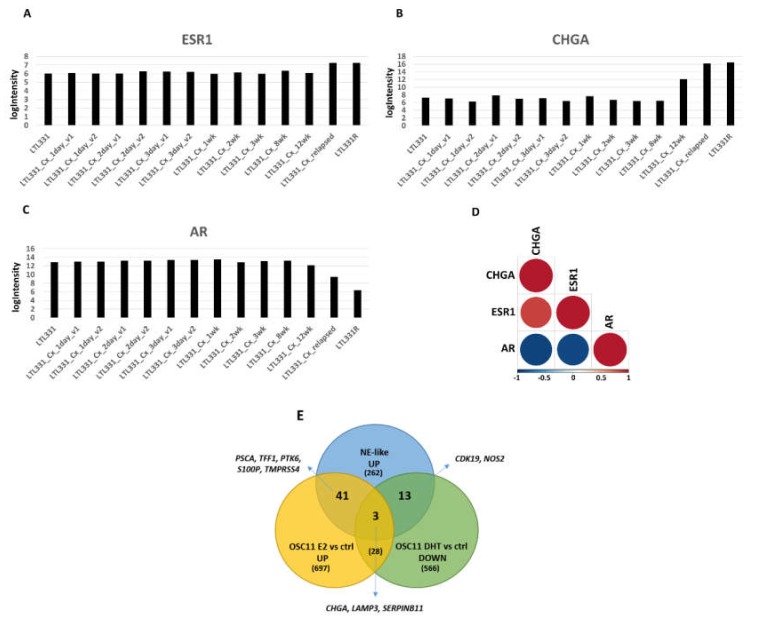
(**A**–**D**) PDX model of NEPC progression evaluated from prostate adenocarcinoma before castration (LTL331) to relapsed neuroendocrine tumor (LTL331R) [16]. ESR1 (**A**) and CHGA (**B**) increased their expression throughout the transdifferentiation process, with concomitant AR down-regulation (**C**). (**D**) Correlation matrix showing a high positive correlation between ESR1 and CHGA (Pearson’s correlation coefficient = 0.86, *p*-value = 7.27 × 10^−5^) and a negative correlation between ESR1 and AR (Pearson’s correlation coefficient = −0.89, p-value = 1.54 × 10^−5^) and between AR and CHGA (Pearson’s correlation coefficient = –0.91, *p*-value = 6.11 × 10^−6^). (**E**) The overlap between up-regulated genes in NE-like vs. AdenoPCa samples and genes up-regulated in OSC11 upon treatment with estradiol (E2) or genes down-regulated in OSC11 upon treatment with DHT, with extrapolation of some genes of interest.

**Figure 5 ijms-21-01078-f005:**
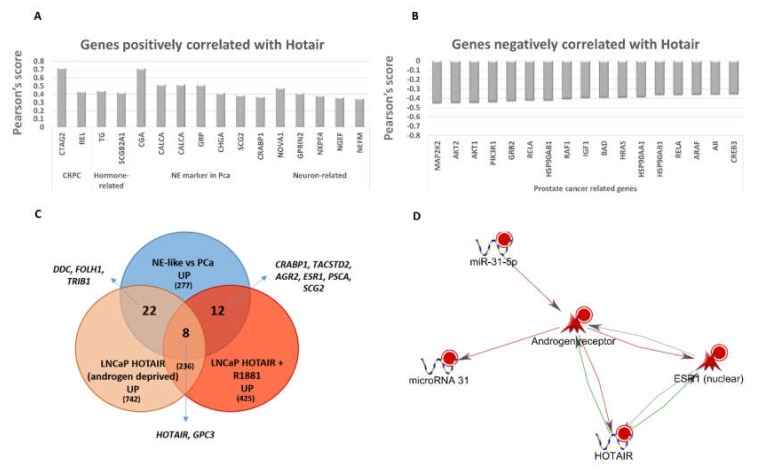
(**A**) Histogram depicting Pearson’s correlation coefficient for a subset of genes whose expression was positively correlated with that of HOTAIR (Pearson score > 0.3 and *p*-value < 0.05). (**B**) Histogram depicting Pearson’s correlation coefficient for a subset of prostate cancer-related genes whose expression was negatively correlated with that of HOTAIR (Pearson score < −0.3 and *p*-value < 0.05). (**C**) Venn diagram of up-regulated genes in NE-like vs. AdenoPCa samples and in hormone-deprived LNCaP cells treated with HOTAIR alone or HOTAIR plus R1881, compared to correspondent controls, with extrapolation of some genes of interest. (**D**) Direct interaction network between AR, ESR1, HOTAIR and miR-31-5p. Red and green edges represent inhibition and activation mechanisms, respectively. Grey edges depict undefined/unknown regulations.

**Figure 6 ijms-21-01078-f006:**
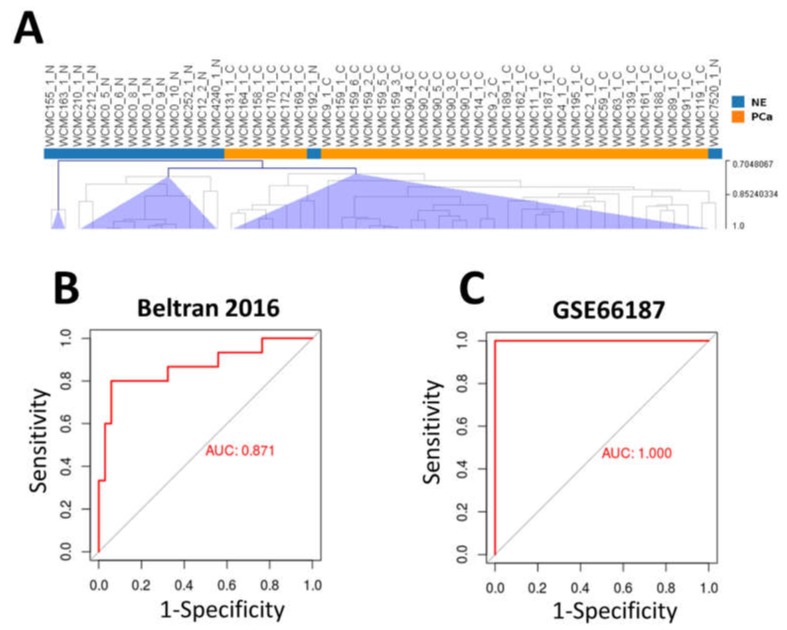
(**A**) Unsupervised hierarchical clustering on 15 CRPC-NE and 34 CRPC-Adeno samples from Beltran et al. [4], using the list of up- and down-regulated genes identified in our cohort of NE-like vs. AdenoPCa samples. Pearson correlation as distance metric and average linkage method were used. (**B**,**C**) Receiver operating characteristic (ROC) curve of the score tested on Beltran dataset (**B**) and on GSE66187 dataset (**C**).

**Table 1 ijms-21-01078-t001:** Distribution of clinical features in NE-like vs. AdenoPCa samples.

	NE-like (*n* = 22)	AdenoPCa (*n* = 32)	Fisher’s Exact Test *p*-Value
Biochemical relapse	12 (55%)	15 (47%)	0.78
Gleason Score			0.02
6	3 (14%)	8 (25%)	
7	11 (50%)	22 (69%)	
8–9	8 (36%)	2 (6%)	
Grade			0.067
2	12 (55%)	26 (81%)	
3	10 (45%)	6 (19%)	
Age at surgery	65.91 ± 5.1	65.86 ± 5.08	0.5
Initial PSA	14.2 ± 15.8	12.9 ± 12.98	0.37

**Table 2 ijms-21-01078-t002:** The number (#) of differentially expressed pcRNAs, ncRNAs and miRNAs in NE-like vs. AdenoPCa samples.

	# pcRNAs	# lncRNAs	# miRNAs
**UP**	295	24	6
**DOWN**	206	15	1

**Table 3 ijms-21-01078-t003:** List of proteins with five or more direct connections (defined as hubs) between up- or down-regulated features in NE-like vs. AdenoPCa samples. Arrows indicate up-regulated (↑) and down-regulated (↓) genes, respectively.

NAME	EDGES
ESR1 (NUCLEAR) ↑	88
ANDROGEN RECEPTOR ↓	82
TFCP2L1 ↑	46
RUNX2 ↑	30
CDX2 ↑	28
C-MYB ↑	16
DNMT3A ↑	15
KLF5 ↑	14
MTOR ↓	13
NKX2-1 ↑	12
BCL-3 ↑	9
MIR-200A-3P ↑	9
MIR-200B-3P ↑	9
C-REL ↑	8
TAF7L ↓	8
ASCL2 ↑	7
NR6A1↑	7
MECOM ↑	7
BATF ↑	6
LDHA ↓	6
MALAT1 ↑	6
MIR-99A-5P ↓	6
HOTAIR ↑	5
IGF-1 ↓	5

**Table 4 ijms-21-01078-t004:** List of known drug target proteins, together with correspondent compounds, that were up- or down-regulated in NE-like vs. AdenoPCa samples.

UP-Regulated Genes	DOWN-Regulated Genes
**HCAR3** (Nicotinic acid) **CXCR2** (Elubrixin, Navarixin, Reparixin, SB332235) **KCNQ1** (Azimilide, Indapamide) **KCND2** (Dalfampridine) **REL** (Apilimod) **AMD1** (N(1),N(11)-Diethylnorspermine) **DDC** (Carbidopa) **ESR1** (Afimoxifene, Clomifene, Diethylstilbestrol, Megestrol, Raloxifene) **NOS2** (GW274150) **ADRA2C** (Apraclonidine, Azepexole, Besipirdine, Brimonidine, Deriglidole, Dipivefrine, Efaroxan, Fipamezole, Guanethidine, Idazoxan, Lusaperidone, Mianserin, Naphazoline, OPC28326, Piperoxan, Tramazoline, Yohimbine) **LPL** (Gemfibrozil, Ibrolipim) **GPNMB** (Glembatumumab vedotin) **CACNA1H** (Mibefradil, Sipatrigine) **GPC3** (Codrituzumab)	**PPIE** (Ciclosporin) **S1PR1** (Fingolimod) **CA3** (Acetazolamide) **KCNMA1** (Cromoglicic acid, Diazoxide, Hydroflumethiazide, Trichlormethiazide) **AR** (Androstanolone, Bicalutamide, BMS564929, Cyproterone acetate, Dehydroepiandrosterone, Diethylstilbestrol, Drospirenone, Finasteride, Flutamide, HE3235, LGD-2226, Megestrol O-acetate, Metandienone, Methyltestosterone, Osaterone, Oxandrolone, Oxendolone, RU58841, Silibinin, Testosterone, Zanoterone) **MTOR** (AZD8055, Everolimus, Ridaforolimus, Sirolimus, Temsirolimus) **PTGS1** (Acemetacin, Aspirin, Benoxaprofen, Bromfenac, butibufen, Celecoxib, Diflunisal, Droxicam, Etodolac, Fenbufen, Fenflumizol, Fenoprofen, Ibufenac, Ibuprofen, Indoprofen, Ketorolac, Lornoxicam, Loxoprofen, Meclofenamic acid, Meloxicam, Mesalazine, Nabumetone, NCX701, Nitroaspirin, Oxaprozin, Parsalmide, Phenacetin, Piroxicam, Salsalate, Sulindac, Tarenflurbil, Tenidap, Tenoxicam, Tiaprofenic acid, Timegadine, Tolfenamic acid, Tolmetin, Zaltoprofen, Zomepirac) **EPHX2** (AR9281)

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
