# Peer review of "Gene Expression Signature Predictive of Neuroendocrine Transformation in Prostate Adenocarcinoma"

_ijms, 2020, doi:10.3390/ijms21031078_

Round 1

Reviewer 1 Report

Authors profiled 54 PCa samples with more than 10-years follow-up for gene and miRNA expression, generated gene expression signature to predict NEPC and tested on independently published datasets. This study identified molecular features (protein coding, long non-coding and microRNAs), at the time of surgery, that may anticipate the NE transformation process of prostate adenocarcinoma.

There are few things that authors may need to clarify.

Increased ESR1 expression in CRPC and NEPC has been reported by other researchers but its role in NEPC differentiation is not clear yet. Authors described more details in cross-talk between AR and ESR1 signaling. Genes regulated by both receptor partially overlapped. Can authors further dissect the potential roles of each set of genes ( AR specific, ESR1 specific and shared) in cancer progression and NEPC differentiation?  NEPC signature has been reported by other groups. How is the signature constructed in this study compared with the previous reported signature? 'ESR1 and AR were among the statistically significant transcription factors for up-regulated genes, while ESR2 and AR for the down-regulated genes.'   - This paragraph has been repeated few times. How does AR involved in opposite conditions? Please ne more specific to avoid the confusion.

Author Response

Major point 1: Increased ESR1 expression in CRPC and NEPC has been reported by other researchers but its role in NEPC differentiation is not clear yet. Authors described more details in cross-talk between AR and ESR1 signaling. Genes regulated by both receptor partially overlapped. Can authors further dissect the potential roles of each set of genes (AR specific, ESR1 specific and shared) in cancer progression and NEPC differentiation? 

Response to Major point 1: Common AR and ESR1 targets, concomitantly down-regulated in NE-like prostate tumors, included LDHA, already described in prostate cancer cells and activated by AR. No studies in PCa were available on the interaction between LDHA and ESR1. Interestingly, among the common up-regulated targets, UGT2B15 and HOTAIR were described to be inhibited by AR in prostate tumors and activated by ESR1.

AR specific targets were mainly involved in sulfur compound biosynthetic process, response to lipid, response to steroid hormone, regulation of ion homeostasis, signal transduction, epithelial cell development. The majority of them has already been described in PCa and most of the down-regulated ones in NE-like tumors were indeed activated by AR.

ESR1 specific targets were mainly involved in T-helper cell differentiation, regulation of hydrolase activity, response to hormone, negative regulation of cell proliferation, regulation of neuron death, cell differentiation. Among ESR1 specific targets that were up-regulated in NE-like tumors, only TFF1 has already been studied in PCa, where it is activated by the estrogen receptor. The other up- or down-regulated targets have not been studied in PCa yet.

Supplementary table 2 contains the list of common and specific up/downregulated targets and the interaction with ESR1 and/or AR, if known. A paragraph has been added in the discussion to describe the potential roles of each set of genes.

Major point 2: NEPC signature has been reported by other groups. How is the signature constructed in this study compared with the previous reported signature?

Response to Major point 2: The main difference between our signature and the previously reported ones consisted in the biological samples used to derive the set of genes able to discriminate adenocarcinomas from neuroendocrine transdifferentiated tumors.

In our study, we started from primary prostate adenocarcinomas, as assessed by an experienced pathologist. On the other hand, the already published studies used metastases from neuroendocrine transdifferentiated prostate cancer, treatment-emergent small cell neuroendocrine prostate cancer (Beltran et al., 2016, Aggarwal et al., 2018), or pure NE tumors (Beltran et al., 2011).

In the study by Beltran et al., 2016, which represents the existing dataset with the largest number of NE transdifferentiated prostate samples, a molecular classifier composed by 70 genes was developed by exploiting expression data of genes prioritized by genomic, transcriptomic or epigenomic status.

Concerning the methodologies used to derive gene signatures in bioinformatics studies, Tsai and colleagues (2017) started from published datasets including adenocarcinomas with NE differentiation, small cell prostate carcinomas, and large cell neuroendocrine carcinomas, using an outlier-based meta-analysis approach.

Aggarwal and colleagues (2018), developed a gene expression signature of t-SCNC, with high internal accuracy on leave-pair-out cross-validation, that was applied to three external mCRPC data sets and correctly categorized neuroendocrine tumors. No detailed information on the methodology was available.

In the study by Cheng et al. (2019), clustering and differential expression analysis was performed to derive a NEPCa signature.

Alshalalfa and colleagues (2019) applied a logistic regression model with ridge penalty, a more appropriate method to build a classifier that was also used in our study. They derived a 212-gene signature to identify treatment-naïve primary prostatic tumors that were molecularly analogous to prostatic small cell neuroendocrine carcinoma. 4 genes were in common between our 8-gene signature and Alshalalfa’s 212-gene signature (SLC35D3, NEFM, SYT4, SMPD3). This result has been added in our paper to corroborate the robustness of our classifier. However, their dataset is not publicly available.

We believe that our modeling approach, that starts from primary prostate adenocarcinomas with 10-years of follow-up, combines results derived from other studies to build a robust classifier and then validates the classifier on neuroendocrine transdifferentiated tumors, adds novelty in this field and yields to robust results.

We have added these observations in the discussion.

Major point 3: 'ESR1 and AR were among the statistically significant transcription factors for up-regulated genes, while ESR2 and AR for the down-regulated genes.'   - This paragraph has been repeated few times. How does AR involved in opposite conditions? Please ne more specific to avoid the confusion.

Response to Major point 3: AR can act both as transcriptional activator or repressor, depending on specific DNA binding elements or specific coactivator and corepressor protein complexes. This peculiarity was highlighted by the fact that we found many up- or down-regulated genes among AR transcriptional targets.

We added a sentence in the Results section to avoid the confusion.

Reviewer 2 Report

The authors profiled in their work prostate cancer tumors on mRNA and non-coding RNA expression levels after grouping them into adenocarcinoma and neuroendocrine prostate cancer. Among others the aim was to better understand the underlying disease biology in NEPC and to find predictors for accurate classification of NEPC and AdenoPCa groups with the parameters measured here. The question the authors raise is highly relevant to the field because of a likely increase of NEPC pateints in the future. A better understanding of NEPC and an early and low error-rate prediction of NEPC will be necessary for better patient outcome in the future. Among others novel points in this work are that MALAT1 and HOTAIR, two lnc-RNAs, as well as the estrogen receptor signaling are being upregulated in NEPC compared to AdenoPCa tumors. Further, a classification model is presented to group NEPC vs AdenoPCa by unsupervised hierachical clustering.

Major points:

Overall the work is comprehensive and show new aspects in NEPC, however a more vigorous proof that MALAT1 and HOTAIR are higher expressed in NEPC compared to AdenoPCa is necessary. Various data are open to public and a correlation of the findings here to other findings in the field are possible – for example NEPC data in the Weill Cornell Medicine (WCM) data set mentioned here in Varune Rohan Ramnarine et al. 2018. 

In addition, it would be interesting to test the WCM dataset for the prediction model shown in figure 6. 

It is interesting that the estrogen receptor signaling pathway is stressed here as a potential upregulated gene signature in NEPC. However, the increase of gene transcription of ESR1 is rather small as shown in Figure 4A. To give more significance to the statement that estrogen receptor signaling is stronger in NEPC than in AdenoPCa tumors the gene expression differences between those groups be shown for ESR1 expression differences together with absolute gene expression levels. Although, there can be a 10-fold difference in gene expression the gene still can be expressed at a very low level. Thus it would be worth to see the overall protein abundance of ESR1 in NEPC (in public data for example). 

Minor points:

The classified groups shown in Figure 2D (between NEPCs vs AdenoPCa) might have another subset within the NEPC group (Fig2D) – the left ten samples or so – it could be worth looking for a stronger NEPC subset within the NEPC group.

As follow up data of the patients studied here exists, it would be interesting to know if the patients classified in the NEPC group have/had worse outcome than the patients in the other group – or if they develop resistance to androgen deprivation therapies.

Figure 2A: Please highlight gene names at dots of the top10 or top 20 DEGs – it would be interesting to see the highly DEGs – or for example use a cutoff for the gene names at log2foldchange absolute values of 1.

Figure 2D: What is the scale/ unit? Is it the mean-z value?

Figure 3A: Please add a multiple testing corrected method in addition (or instead) to the p-values for the pathways identified. Please mention if those pathways selected are the top hits, or if it is a subset selected from a list. If it is a subset from a list – please provide the complete list.

Figure 4E/ 5C: please provide the numbers of the non-overlapping parts of the diagram. How many genes are within the other parts of the diagram?

Round 2

Reviewer 1 Report

Authors convincingly addressed most of the reviewers' concerns. 

Reviewer 2 Report

The authors clearly addressed all the points and improved the manuscript. Excellent work!